# Incorporating Dynamic Traffic Distribution into Pavement Maintenance Optimization Model

**Xinhua Mao [1,2,\*], Changwei Yuan [1] and Jiahua Gan [3]**

[1]  School of Economics and Management, Chang'an University, Xi'an 710064, China; changwei@chd.edu.cn
[2]  Department of Civil and Environmental Engineering, University of Waterloo, Waterloo,
   ON N2L 3G1, Canada
[3]  Transport Planning and Research Institute, Ministry of Transport, Beijing 100028, China; ganjh@tpri.org.cn
[\*]  Correspondence: mxinhua@uwaterloo.ca

**Abstract:** An optimal pavement maintenance strategy can keep the pavement performance at a high level under budget constraint. However, the impact of changes in traffic distribution caused by maintenance actions on user costs is rarely investigated in existing approaches. This research aims to solve the optimization of pavement maintenance strategy using a multi-stage dynamic programming model combined with the stochastic user equilibrium model, which can simulate the dynamic traffic distribution in the life cycle. To deal with the proposed model, a heuristic iterative algorithm is put forward. Ultimately, a hypothetical network is established to test the model and algorithm. The testing results prove that the proposed framework has an advantage in assessing user costs comprehensively and can provide an effective and optimal pavement maintenance strategy in a 30-year life cycle, which improves the efficiency of budget and pavement conditions. Additionally, this research provides quantitative evidence of interdependency in a road network, i.e., pavement maintenance actions on links can interfere with the user costs and traffic flow distribution in the whole network, which should be taken into account in pavement maintenance decision-making.

**Keywords:** pavement maintenance optimization; dynamic traffic distribution; stochastic user equilibrium; life cycle; maintenance cost; user cost; heuristic iterative algorithm

## 1. Introduction

Pavement provides indispensable physical facilities for transport activities, which needs periodic maintenance actions to ensure high levels of performance. Unfortunately, the available budget cannot afford to maintain all the roads simultaneously in a network. Hence, an effective maintenance plan is necessary for road managers. In view of this, many studies have been carried out focusing on pavement maintenance management, which is defined as a resource allocation strategy for pavement maintenance of road segments in one or multiple time periods [1]. The main purpose of the strategy is to reduce user costs, prolong pavement life span, and preserve the value of road assets [2]. The decision results are always based on the tradeoff between user costs and agency costs [3]. The former includes fuel consumption, travel delay, vehicle maintenance cost, etc., while the latter is mainly associated with maintenance costs.

The life cycle of the pavement system consists of multiple periods, and every period includes two stages, i.e., service stage and maintenance stage. In the former stage, pavements provide normal service to users without maintenance actions, while, in the latter stage, pavements are given maintenance actions. Figure 1 shows that in a pavement life cycle with $T$ periods, $t_0$ is the service stage, and $t_1$–$t_0$ is the maintenance stage of the first period $t_1$. Traffic flow distribution varies dynamically on the basis of pavement performance loss in the service stage, while, in the maintenance stage, the bottleneck

formed by maintenance actions reduces traffic capacities, changes traffic distribution and increases traffic delay, which affects user costs. As a result, the interactive relationship between user costs and pavement performance, as well as maintenance actions, should be investigated when maintenance strategies are made.

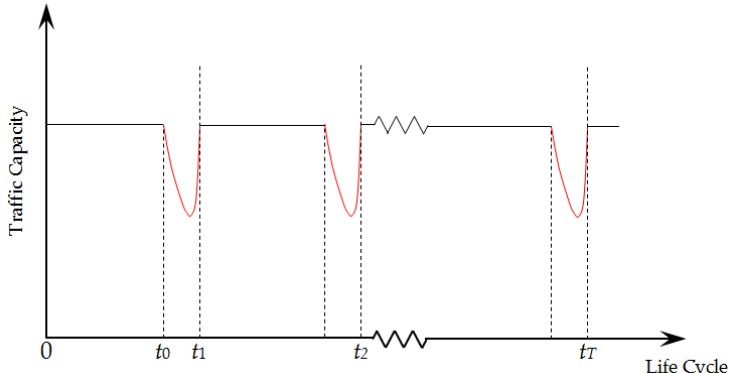

**Figure 1.** Changes of traffic capacities in the pavement life cycle.

The existing approaches associated with pavement maintenance optimization can be classified into two categories, i.e., top-down and bottom-up. The former assumes that facilities have homogenous characteristics such as pavement type, performance deterioration, and so on [2,4], while the latter takes different attributes of all facilities into consideration [5,6]. However, the two groups of approaches both assume that the facilities are independent, and they often do not account for the dynamic traffic distribution during the life cycle. Comprehensive impact of the interactive relationship between user costs and pavement performance, as well as maintenance actions on pavement maintenance decision-making, has not been completely investigated.

To fill this gap, we will calculate the life cycle user costs by simulating the dynamic traffic distribution in both service and maintenance stages in every period using stochastic user equilibrium (SUE) model and establish the pavement maintenance optimization model subject to SUE. In particular, pavement maintenance decision is a stochastic dynamic process with the uncertainty of pavement performance transfer, which is usually formulated as an optimization problem [7]. In addition, as a traffic flow assignment method, SUE can simulate the distribution of traffic flow and traffic delay at different times in a road network dynamically [8]. For better decision-making of pavement maintenance, this research extends the approach from three aspects, namely, (i) formulating cost functions based on dynamic traffic simulation, (ii) proposing pavement maintenance optimization model with life cycle costs (LCC) minimization, and (iii) developing a solution algorithm.

This research makes the following two contributions. First, we precisely and comprehensively identify the life cycle user costs generated in all stages of every period. Second, we propose a framework to make optimal pavement maintenance strategies considering dynamic traffic distribution in every time period.

The remainder of this paper is organized as follows. Section 2 reviews the modeling and algorithm of pavement maintenance optimization. Section 3 formulates the problem, gives assumptions, proposes the decision-making models, and develops the solution algorithm. Section 4 develops a numerical example of a hypothetical network and presents the testing results. Section 5 discusses the key findings of the results. The conclusions are drawn in Section 6.

## 2. Literature Review

Optimal pavement maintenance plans mainly solve the resource allocations for maintenance actions of multiple pavements. Linear programming [9], nonlinear programming [10], and mixed integer programming [11] were usually adopted to solve the problem formulated as combinatorial

optimization by many researchers. For example, Ouyang employed a nonlinear programming model setting the minimum sum of maintenance costs and user costs as the objective function to obtain the optimal pavement resurfacing plan [12]. As well as the objective function developed by Ouyang, maximum pavement performance [13], minimum International Roughness Index (IRI) [14], minimum pavement salvage values [15], minimum greenhouse gas emissions [16], and maximum user benefits [17] were also considered as objective functions in existing literature, which makes the decision problem capable of being solved by multi-objective optimal models [17]. The above optimal approaches are applicable to the single-stage decision problem but have limitations in solving multi-stage decision problems. To fill this gap, many other researchers formulated the problem as a constrained Markov Decision Process (MDP), which was solved using stochastic programming models [18]. For instance, Medury and Madanat established the pavement maintenance mode selection set using a backward recursive function, based on which an MDP-based optimization model was put forward to obtain the multi-stage pavement maintenance strategy [19]. Khan et al. proposed an approximate dynamic programming model for the optimal decision problem of multi-stage pavement maintenance [20].

With the increase of the complexity of pavement maintenance decision-making modeling, the ability of the algorithm to obtain global or local optimal solutions is more and more important. Some accurate algorithms such as linear programming algorithm [21], gradient descent algorithm [22], branch and bound algorithm [23], etc. were introduced to solve the single-stage maintenance decision problems with a small feasible region. For multi-stage maintenance decision models, Lagrange multiplier method [16], Lagrange dual method [24], approximation algorithm [12], etc. were employed as solution algorithms. Since pavements have different deterioration characteristics, the convexity of the maintenance decision models varies in different situations. However, the effectiveness of such algorithms depends on the convexity of the models, which makes these algorithms not universally suitable for all models [25]. Hence, heuristic algorithms are gradually applied to solve the complex models of pavement maintenance optimal decision, within which the tabu search approach [26] and genetic algorithm [3] are the two most commonly used heuristic algorithms for pavement maintenance decision models. However, these two algorithms cannot guarantee that the obtained solution is globally optimal and cannot predict the deviation degree between the feasible solution and the optimal solution [27].

Despite the wide range of modeling, it is rare in literature to incorporate dynamic traffic distribution into pavement maintenance optimization models. This research develops an optimal model using a dynamic programming approach subject to SUE model considering the interactive relationship between user costs and traffic distribution. Only in this way can user costs be evaluated accurately and comprehensively in the life cycle. Additionally, a heuristic iterative algorithm is developed to solve the model.

## 3. Methodology

### 3.1. Assumptions

For simplicity, we give the following assumptions:

(1)　The pavement life cycle is divided into periods by year.
(2)　Users have complete information of the road network, which affects users' route choice, i.e., traffic distribution. Traffic distribution depends on pavement performance deterioration in the service stage, while in the maintenance stage, traffic distribution is determined by both pavement performance deterioration and traffic capacity loss.
(3)　Maintenance action is expected to be completed in the shortest possible time.
(4)　Maintenance actions shall be carried out for more than one road simultaneously instead of being one by one.
(5)　Performance of every road is restored to the initial level after a maintenance action.
(6)　Maintenance actions will not block the roads completely, i.e., traffic capacity is not zero under maintenance.

(7)　　Roads in the network are constructed with asphalt pavements.

### 3.2. General SUE Model

According to SUE model, the traffic volume of link *s* in a road network is $Q_s$, which can be calculated by Equation (1) in [28].

$$\sum_{k=1}^{K_w} \sum_{i=1}^{n} \sum_{j=1}^{m} h_k^{wij} \cdot \delta_s^{wkij} = \sum_{i=1}^{n} \sum_{j=1}^{m} Q_{sij} = Q_s, \forall s \in N, \forall w \in W \tag{1}$$

where $Q_{sij}$ is the traffic volume of vehicle class *i* in axle weight group *j* on link *s*, $s = 1, 2, \cdots, N$, $i = 1, 2, \cdots, n, j = 1, 2, \cdots, m$; *w* is an origin-destination (OD) pair, which belongs to OD set *W*, $w \in W$; *k* is a path linking OD pair *w*, $k \in K_w$; $K_w$ is a set of paths; $q^w$ is traffic demand of OD pair *w*; $\delta_s^{wkij}$ is a logical variable, which is 1, if link *s* lies on path *k* linking OD pair *w* and 0, otherwise; $h_k^{wij}$ is the traffic volume of vehicle class *i* in axle weight group *j* on path *k* linking OD pair *w*, which should be subject to

$$\sum_{k=1}^{K_w} \sum_{i=1}^{n} \sum_{j=1}^{m} h_k^{wij} = q^w, \forall w \in W \tag{2}$$

where $q^w$ is the traffic need of OD pair *w*.

### 3.3. Pavement Deterioration Model

In this research, we focus on asphalt pavements, which are most widely used with the advantages of good fatigue durability, crack resistance and water stability [29–32]. Asphalt pavement performance is measured by the Present Serviceability Index (PSI) [33]. Traffic loads generated by all vehicles to pavements are measured by the equivalent number of standard axle loads (ESALs) [34]. The function of PSI depending on ESALs can be formulated as Equation (3).

$$PSI = \frac{P_0}{P_0 - P_c} \cdot \alpha \cdot ESALs^\beta \tag{3}$$

where $P_0$ is PSI level at the beginning of a time period; $P_c$ is the minimum threshold of PSI level; $\alpha$ and $\beta$ are calibration coefficients; *ESALs* is the cumulative ESAL during a time period, which is computed by Equation (4).

$$ESAL = \sum_{i=1}^{n} \sum_{j=1}^{m} Q_{ij} \cdot LEF_{ij} \tag{4}$$

where $Q_{ij}$ is the traffic volume of vehicle class *i* in axle weight group *j*, $i = 1, 2, \cdots, n, j = 1, 2, \cdots, m$; $LEF_{ij}$ is load factor determined by vehicle class and axle weight group.

### 3.4. Maintenance Cost Function

We assume that every pavement will be restored to the initial PSI level after maintenance actions, i.e., at the start of the next period. Pavement maintenance costs are determined by the loss of PSI. The life cycle is assumed to be divided into several periods by year, and then the maintenance cost vector $M^t$ of all links in the *t*th year is

$$M^t = \left[ M_1^t, M_2^t, \cdots, M_s^t \right], \forall s \in N, \forall t \tag{5}$$

$$M_s^t = UC \cdot \left( PSI_{s0}^t - PSI_{sc}^t \right) \cdot L_s \cdot \sigma_s \tag{6}$$

where $M_s{}^t$ is the maintenance costs needed for link *s* in the *t*th year; *UC* is the unit maintenance cost determined by PSI loss; $PSI_{s0}{}^t$ is the PSI level of link *s* at the beginning of the *t*th year; $PSI_{sc}{}^t$ is the terminal PSI level at the end of service stage in the *t*th year; $L_s$ is the length of link *s*; $\sigma_s$ is the number of lanes of link *s*.

### 3.5. Use Cost Function

#### 3.5.1. Travel Time

Users' travel time vector $E^t$ on all links in the $t$th year can be formulated as

$$E^t = [E_1^t, E_2^t, \cdots, E_s^t], \forall s \in N, \forall t \tag{7}$$

$$e_s^t(Q_{ij}, C_s) = \sum_{s=1}^{N} \sum_{i=1}^{n} \sum_{j=1}^{m} \int_0^{Q_{sij}} f_s(Q_{ij}, C_s) d_{Q_{ij}}, \forall s \in N, \forall t \tag{8}$$

$$f_s(Q_{ij}C_s) = t_s^0 \left[ 1 + \theta \left( \frac{Q_{sij}}{C_s} \right)^\xi \right] = \left( \frac{L_s}{v_s^0} \right) \left[ 1 + \theta \left( \frac{Q_{sij}}{C_s} \right)^\xi \right], \forall s \in N, \forall t \tag{9}$$

where $e_s^t(Q_{ij}, C_s)$ is the average travel time per day on link $s$ in the $t$th year; $t_s^0$ is free travel time on link $s$; $v_s^0$ is free travel speed on link $s$; $\theta$ and $\xi$ are two calibration coefficients; $C_s$ is the observed actual traffic capacity of link s, which may have different values in the service stage and maintenance stage.

$$C_s = \begin{cases} C_s^*, & \text{in the service stage} \\ \rho C_s^*, & \text{in the maintenance stage} \end{cases} \tag{10}$$

where $C_s^*$ is the actual traffic capacity of link $s$ in the service stage; $\rho C_s^*$ is the actual traffic capacity of link $s$ in the maintenance stage, where $0 < \rho < 1$, if link $s$ is given the maintenance action, and $\rho = 1$, otherwise.

Accordingly, the total travel time on link $s$ in the $t$th year can be formulated as Equation (11) based on the hypothesis that maintenance actions will be completed in the shortest possible time.

$$E_s^t = (365 - \pi_t) \cdot e_s^t(Q_{ij}, C_s^*) + \pi_t \cdot e_s^t(Q_{ij}, \rho C_s^*), \forall s \in N, \forall t \tag{11}$$

where $\pi_t$ is the shortest time span of maintenance stage in $t$th year, which is calculated as follows.

$$\pi_t = \text{Max}\{\pi_t^1, \pi_t^2, \cdots, \pi_t^s\} \tag{12}$$

where $\pi_t^s$ is the maintenance time of link $s$ in the $t$th year. $\pi_t^s$ has the form shown in Figure 2, which indicates that there is an exponential relationship between the maintenance time of each link and its present PSI, and can be formulated as

$$\pi_t^s = \text{Ceil}\left[ \pi_s^{max} \cdot \left( \frac{PSI_{s0}^t - PSI_{sc}^t}{PSI_{s0}^t} \right)^\eta \right] \tag{13}$$

where $\pi_s^{max}$ is maximal days required for maintenance action of link $s$ when its terminal PSI is 0. $\eta$ is a calibration parameter. Ceil [·] rounds the element to the nearest integers toward infinity.

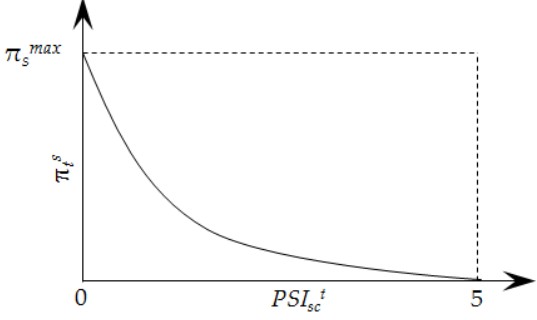

**Figure 2.** Relationship between $PSI_{sc}{}^t$ and $\pi_t^s$.

### 3.5.2. Driving Cost

Driving cost vector $D^t$ of all links in the $t$th year can be formulated as

$$D^t = [D_1^t, D_2^t, \cdots, D_s^t], \forall s \in N, \forall t \tag{14}$$

$$d_s^t(Q_{ij}, PSI_s, C_s) = \sum_{s=1}^{N} \sum_{i=1}^{n} \sum_{j=1}^{m} \int_0^{Q_{sij}} \int_{PSI_{s0}}^{PSI_{sc}} f_s(Q_{ij}, PSI_s, C_s) d_{Q_{ij}} d_{PSI_c}, \forall s \in N, \forall t \tag{15}$$

$$f_s(Q_{ij}, PSI_s, C_s) = \left[ \tau + \gamma v_s^c + \varepsilon(v_s^c)^2 + \epsilon PSI_s + \mu PSI_s^2 \right] \cdot L_s \cdot \sigma_s \tag{16}$$

$$v_s^c = \frac{v_s^0}{\left[ 1 + \theta \left( \frac{Q_{sij}}{C_s} \right)^{\xi} \right]} \tag{17}$$

$$D_s^t = (365 - \pi_t) \cdot d_s^t(Q_{ij}, PSI_s, C_s^*) + \pi_t \cdot e_s^t(Q_{ij}, PSI_s, \rho C_s^*), \forall s \in N, \forall t \tag{18}$$

where $d_s^t\left(Q_{ij}, PSI_s, C_s\right)$ is the average driving costs per day on link $s$ in the $t$th year; $\tau$, $\gamma$, $\varepsilon$, $\epsilon$, and $\mu$ are calibration coefficients.

### 3.5.3. User Cost

Total user costs $U^t$ in $t$th year is determined by travel time and driving costs as Equation (19).

$$U^t = E^t + D, \forall t^t \tag{19}$$

### 3.6. Salvage Value

We use a simple function to evaluate the salvage value of each link as in Reference [17], which is formulated as

$$SV_s = (1 - \frac{LT_s}{ET_s}) \cdot M_s^T, \forall s \tag{20}$$

where $SV_s$ is salvage value of link $s$; $LT_s$ is the left lifetime of link $s$; $ET_s$ is the expected lifetime of link $s$; $M_s^T$ is the maintenance costs of the last maintenance action of link $s$.

### 3.7. Pavement Maintenance Optimization Model with LCC Minimization

$$\text{Minimize LCC} = C_0 + \sum_{t=1}^{T-1} \frac{U^t + M^t A^t}{(1 + \omega)^t} - \sum_{s=1}^{N} \frac{SV_s}{(1 + \omega)^T} \tag{21}$$

Subject to the constraints

$$\sum_{k=1}^{K_w} \sum_{i=1}^{n} \sum_{j=1}^{m} h_k^{wij} = q^w, \forall w \in W \tag{22}$$

$$\sum_{k=1}^{K_w} \sum_{i=1}^{n} \sum_{j=1}^{m} h_k^{wij} \cdot \delta_s^{wkij} = Q_s, \forall s \in N, \forall w \in W \tag{23}$$

$$M^t A^t \leq B, \forall t^t \tag{24}$$

$$PSI_{s0}^t = PSI_{sc}^{t-1} \tag{25}$$

$$h_k^{wij} \geq 0, \forall k \in K_w \tag{26}$$

where, $C_0$ is the construction cost of the road network; $\omega$ is discount rate; $A^t = [a_1^t, a_2^t, \cdots, a_s^t], \forall s \in N$, $a_s^t = 1$, if a maintenance action is provided for link $s$, and 0, otherwise; $B^t$ is the budget vector in the $t$th year.

Equation (21) is the objective function, which ensures that the life cycle costs are minimized; Equation (22) indicates that the traffic demand of a OD pair assigns traffic flows on all paths linking the OD pair; Equation (23) shows that the actual traffic volume on a link is constituted of the traffic

volume of all paths on which the link lies; Equation (24) makes sure that the maintenance costs in each year cannot exceed the budget available; Equation (25) reveals that PSI level of each link at the end of the previous year equals to that at the beginning of the later year; Equation (26) sets the traffic volume between a specific OD pair as a positive value.

### 3.8. Model Solution

The simulation of the optimal model proposed above is conducted by the following heuristic iterative algorithm, which is based on Bellman's principle of optimality, e.g., whatever the initial decisions are, the remaining decisions are only dependent on the state resulting from the first decisions.

Step 1: Initialization. Denote $A^{t*}$ as the optimal decision vector; $g\left(A^{t\prime}\right)$ is a set including the links selected to provide maintenance actions; Links with no maintenance actions belong to $g\left(A^{t\prime\prime}\right)$. Set initial $A^{t*} = [a_1^{t0}, a_2^{t0}, \cdots, a_s^{t0}] = [0, 0, \cdots, 0]$, which indicates that no links are selected. Hence, $g\left(A^{t\prime}\right) = \Phi$.

Step 2: Calculate the traffic flow made up of various vehicle classes of different axle weight groups on each link using OD data. Denote $Q_s^z = \sum_{i=1}^{n} \sum_{j=1}^{m} Q_{sij}^z$ as the traffic volume on link $s$ obtained in $z$th iteration. Calculation of traffic volume is executed as following iterative steps:

Step 2a: Denote $Q_s^{z,p}$ as the traffic volume on link $s$ based on the free travel time $t_{s0}^z$, $p = 1$, $\forall z$. $t_s^z$ is computed by the Bureau of Public Roads (BPR) function.

Step 2b: Calculate the new actual travel time $t_s^z$ on all links using BPR function based on $Q_s^{z,p}$.

Step 2c: Calculate the auxiliary traffic flow $Q_s^{z,p\prime}$ of all links using new actual travel time $t_s^z$.

Step 2d: Adjust traffic volume $Q_s^{z,p+1}$ on all links using Equation (27).

$$Q_s^{z,p+1} = Q_s^{z,p} + \frac{1}{N}(Q_s^{z,p\prime} - Q_s^{z,p}) \tag{27}$$

Step 2e: Let $\left|(Q_s^{z,p+1} - Q_s^{z,p})/Q_s^{z,p+1}\right| < \Omega_1$ as the convergence criterion with an allowable error $\Omega_1 \in \forall > 0$. $Q_s^{z,*} = Q_s^{z,p+1}$ is the optimal result, if $Q_s^{z,p+1}$ meets the convergence criterion. Otherwise, denote $p = p + 1$, return to Step2b, and execute the next iteration till the convergence criterion is met.

Step 3: Employ a penalty function to transform the model into the following optimization model without constraints.

$$\text{Minimize } f(A^t, \chi) = \text{Minimize}\left[C_0 + \sum_{t=1}^{T-1} \frac{U^t + M^t A^t}{(1 + \omega)^t} - \sum_{s=1}^{N} \frac{SV_s}{(1 + \omega)^T} + \chi\left(M^t A^t - B^t\right)^2\right] \tag{28}$$

where, $\chi$ is the penalty factor. $\chi = 0$, if $A^t$ meets Equation (24), $\chi \in \forall > 0$. Set error gap $\Omega_2 \in \forall > 0$ for initial $A^{t*}$.

Step 4: Solve the unconstrained model Minimize $f(A^t, \chi)$ and denote its optimal solution $A^{z,t*} = A^{z,t*}(\chi^z)$, as follows.

$$\text{Minimize } f(A^{z,t*}, \chi) = \text{Minimize}\left[C_0 + \sum_{t=1}^{T-1} \frac{U^t + M^t A^{z,t*}}{(1 + \omega)^t} - \sum_{s=1}^{N} \frac{SV_s}{(1 + \omega)^T} + \chi\left(M^t A^{z,t*} - B^t\right)^2\right] \tag{29}$$

Step 5: Perform the convergence test. If $M^t A^{z,t*} - B^t > \Omega_2$, the iteration stops and obtain the optimal solution $A^{z,t*}$. Otherwise, set $\chi_{z+1} = \varsigma\chi_z$, $\varsigma \in [5,10]$, $z = z + 1$, and repeat the procedure from steps 2 to 5.

## 4. Numerical Examples

### 4.1. Hypothetical Network and Basic Data

We use a road network shown in Figure 3 to test the model. The network is constituted of 11 links with bituminous pavements. For simplicity, we assume that all links in the network have the

same characteristics, i.e., the length of each link is 2 km, ideal traffic capacity is 15,000 vehicles per day, design speed is 100 km/h, initial PSI is set as 5.0. Each link has three 3.75 m lanes per direction. The life cycle is divided into 30 periods by year.

We classify vehicles into 13 groups using the classification method from Federal Highway Administration (FHWA), and axle loads of vehicles are divided into 10 groups as in Reference [35]. Table 1 displays the traffic demands of 13 different vehicle groups from O to D.

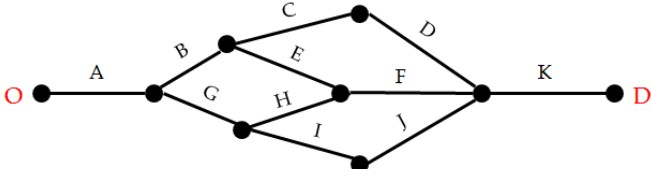

**Figure 3.** Hypothetical road network.

**Table 1.** Traffic demands of 13 different vehicle groups.

| Vehicle Class | O → D (Vehicles/Day) |
|---|---|
| 1 | 4 |
| 2 | 7468 |
| 3 | 176 |
| 4 | 1764 |
| 5 | 1323 |
| 6 | 829 |
| 7 | 241 |
| 8 | 1441 |
| 9 | 1882 |
| 10 | 1452 |
| 11 | 47 |
| 12 | 56 |
| 13 | 13 |

The upper level of budget every year is US$ 45,000. For maintenance costs, we use the value of unit maintenance cost in Reference [28], which includes material expense, transportation fee, labor cost, tax, and so on. The other parameters are set as in Table 2.

**Table 2.** Parameter settings.

| Parameters | Value | Parameters | Value | Parameters | Value |
|---|---|---|---|---|---|
| $\alpha$ | $-4.125 \times 10^{-12}$ | $\rho$ | 0.73 | $\gamma$ | $-1.532$ |
| $\beta$ | $-0.865$ | $\pi_s^{max}$ | 6 | $\epsilon$ | 0.124 |
| $\theta$ | 0.48 | $\eta$ | 0.52 | $\epsilon$ | $-0.367$ |
| $\xi$ | 2.82 | $\tau$ | 31.25 | $\mu$ | 0.005 |

*4.2. Results*

MATLAB R2018b (version 11.4) software was utilized to execute the program based on Microsoft Windows. In the executing procedure, we set the parameters as $\Omega_1 = 10^{-4}$, $\Omega_2 = 10^{-4}$, $\chi = 10$, $z = 50$, $p = 30$.

4.2.1. Optimal Maintenance Strategy

The outcome of the model is an optimal maintenance strategy, i.e., a set of links selected to provide maintenance actions in every period, shown in Figure 4. Maintenance budget can only cover the pavement maintenance expenditures of five links in each period. Since link A and link K are two arteries in the network, which have the highest traffic volume of 16,696 vehicles per day, they have a

great PSI loss and need a maintenance action in each period to reduce user costs. However, Link I and Link J both have the lowest traffic volume, which leads to a minimum PSI loss making the longest time span between two maintenance actions of four years maximum. In the first and some other periods, the worst links are given to the priority of maintenance, but the worst-first strategy is not applicable to all period. In 16th and some other periods, a best-first strategy is adopted, i.e., the best links are given to the priority of maintenance.

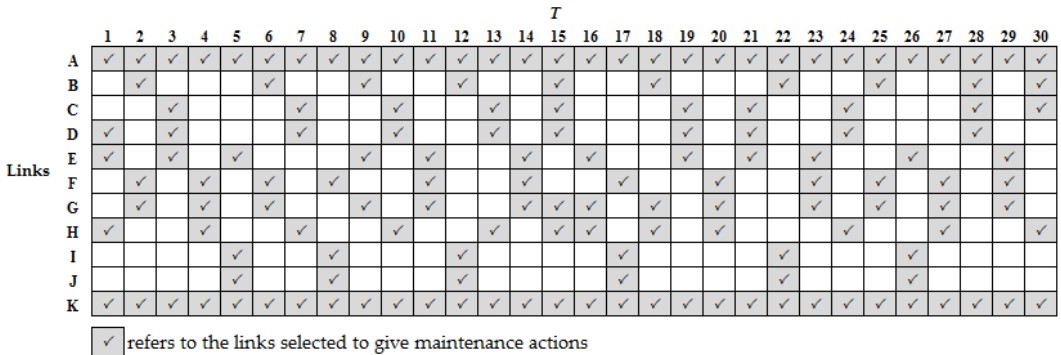

**Figure 4.** Optimal maintenance strategy of every period.

Figure 5 displays the PSI level of each link at the end of every period. Performance of every road is restored to the initial level by a maintenance action. The optimal maintenance strategy gets a flexible PSI threshold triggering maintenance actions with 2.51 minimum and 3.81 maximum, for example, Link I is given a pavement maintenance action under its PSI level of 2.51 in fifth year, while Link G is provided a maintenance action when its PSI level is 3.81 in the 18th year.

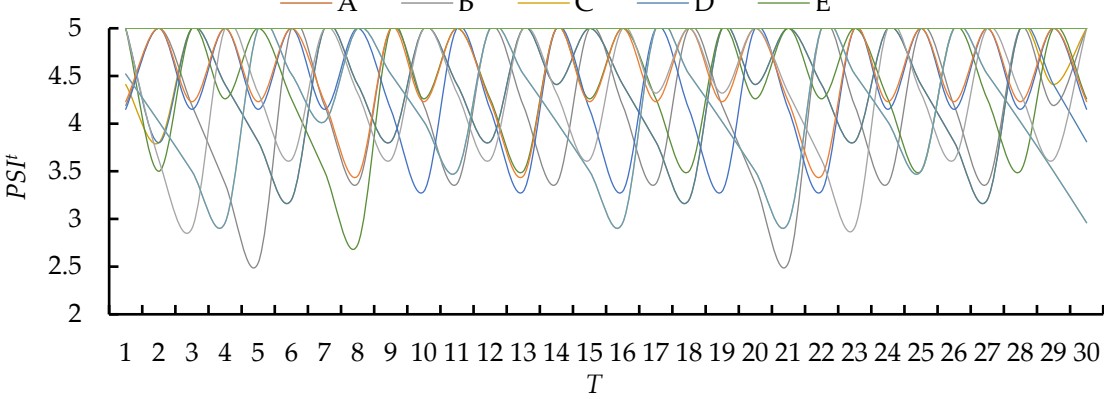

**Figure 5.** $PSI^t$ of each link at the end of every period.

### 4.2.2. Maintenance Cost

We calculate the total maintenance costs in every period respectively, shown in Figure 6, which indicates that the average maintenance costs are US$ 41,057 per period. It consumes the highest expenditure in the ninth and 17th periods and the lowest expenditure in the 24th period. Since maintenance costs are mainly affected by PSI loss in this research and every link has a different PSI loss in each period, there are fluctuations in maintenance costs among the 30 periods.

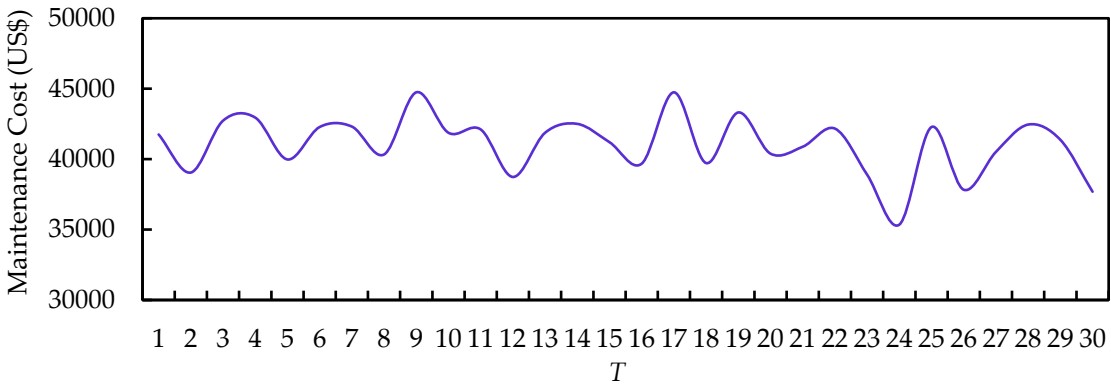

**Figure 6.** Maintenance costs in every period.

### 4.2.3. User Cost

1. Travel Time

Total travel time of each period is composed of two parts: Travel time in the service stage and travel time in the maintenance stage. Because of different time spans of maintenance stage affected by PSI loss, travel time both in the service stage and maintenance stage vary in periods, shown in Figure 7. The 19th period has the maximum travel time of 862,424.86 h, while the eighth period has the minimum travel time of 851,824.85 h.

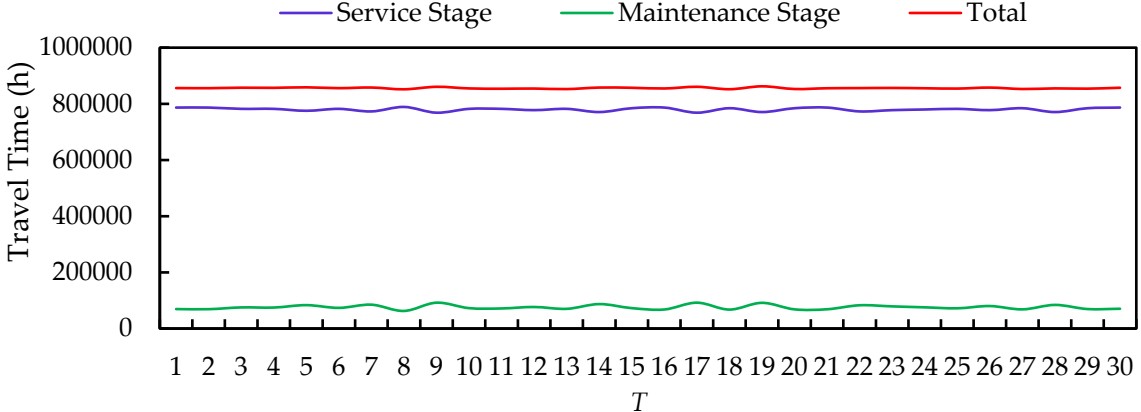

**Figure 7.** Travel time in every period.

2. Driving cost

Like travel time, driving costs also include two parts. Using Equations (14)–(18), we obtain driving costs in each period shown in Figure 8. The 28th period has the maximum driving costs of US $9,882,458, while the 30th period has the minimum travel time of US $9,656,192.

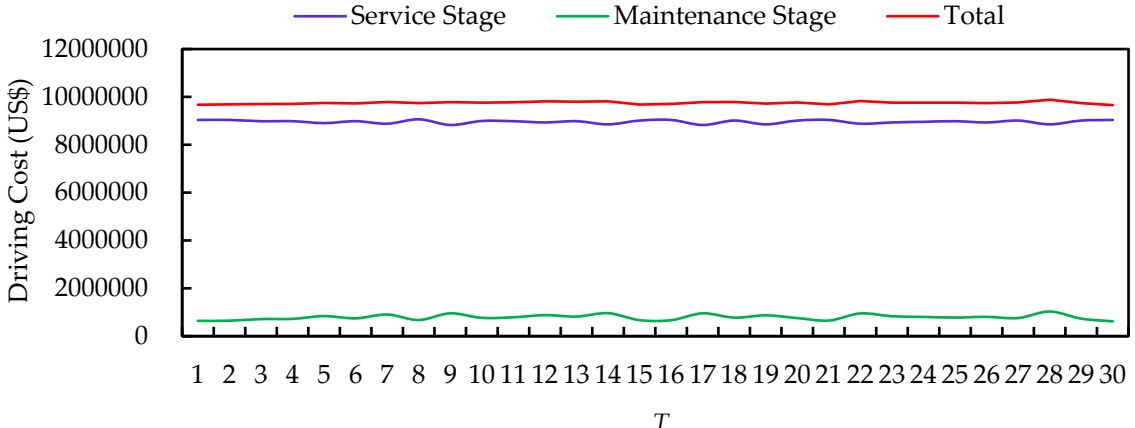

**Figure 8.** Driving costs in every period.

## 5. Discussion

This research explores an approach to make a life cycle optimal pavement maintenance strategy considering the network effect. We discuss our key findings as follows.

### 5.1. Maintenance Strategy

Compared to worst-first strategy and best-first strategy, optimal pavement maintenance strategy refers to using optimization methods to generate a schedule of maintenance actions in one or multiple time periods [36], which can ensure the efficient use of maintenance resources [37]. A maintenance strategy is generated by a multi-stage dynamic programming model in this research, which is a set of plans in all time periods. However, it is found that in some single time period, worst-first strategy or best-first strategy is still adopted, such as worst-first strategy in the first period and best-first strategy in the 16th period. To some extent, optimal pavement maintenance strategy is a mixed strategy under the constraint of budget, which is similar to a previous study [6]. Additionally, to our knowledge, the method proposed in this research is an optimization model under the efficiency of capital allocation, which is also applicable in the optimal investment issue of highway projects and other kinds of infrastructures, e.g., subway, railway, airport.

### 5.2. Interdenpendency in Road Network

Interdependency widely exists in infrastructure systems, which is defined as the state of one infrastructure system can affect the state of others. However, it is rare in literature to capture the interdependency in pavement maintenance strategy. This research provides quantitative evidence that in a road network, pavement maintenance actions on some links can interfere with the user costs and traffic flow distribution in the whole network, which is in line with the study of Uchida and Kagaya [38]. We calculate the traffic flow, travel time and driving cost in the service stage and maintenance stage respectively taking the 10th period for example. Changes of the three parameters caused by maintenance actions of five links are shown in Figure 9, which indicate that maintenance actions of five links divert the traffic flows by 11%, increase the average travel time by 44%, and increase the average driving costs by 31%.

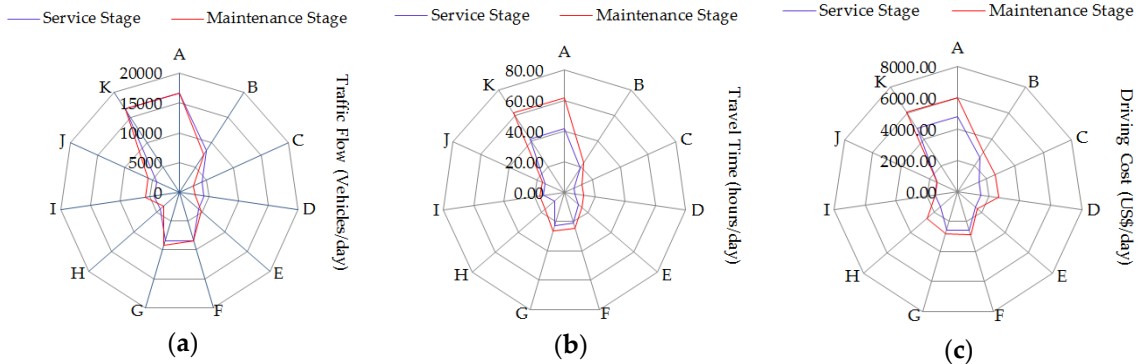

**Figure 9.** Comparison of three parameters in two stages in the 10th period. (**a**) Traffic flow; (**b**) Average travel time; (**c**) Average driving costs.

### 5.3. Sensitivity Analysis

$\rho$ is a parameter that determines the actual traffic capacity of links in the maintenance stage, e.g., $0 < \rho < 1$, if link $s$ is given maintenance action, and $\rho = 1$, otherwise. The value of $\rho$ is affected by the maintenance methods, which has an influence on the decision results. In this section, we discuss the impact on life cycle user costs by different values of $\rho$. Figure 10 displays the sensitivity analysis results of $\rho$.

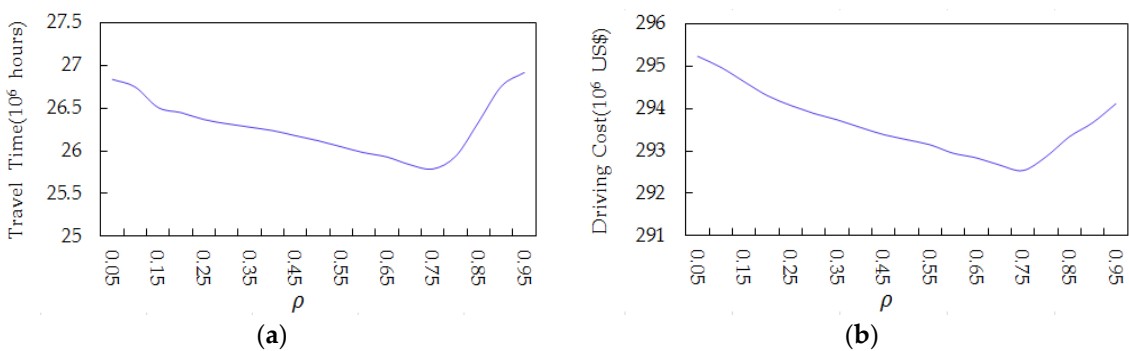

**Figure 10.** Sensitivity analysis of $\rho$. (**a**) Impact of $\rho$ on life cycle travel time; (**b**) Impact of $\rho$ on life cycle driving costs.

From Figure 10, the fluctuations in travel time and driving costs are both an inverted U-shaped curve. $\rho = 0.73$ is the minimum point for both travel time and driving costs. A lower value of $\rho$ means lower traffic capacity, which brings about lower traffic flow velocity. A higher value of $\rho$ means higher traffic capacity, which can increase traffic flow velocity but extends the time period of maintenance stage. When $\rho < 0.73$, the traffic flow velocity is the main reason for user costs. User costs are mainly affected by long maintenance period when $\rho > 0.73$. Hence, the optimal value setting of $\rho$ is the tradeoff between traffic flow velocity and the time period of the maintenance stage.

## 6. Conclusions

An optimal pavement maintenance strategy aims to ensure that the pavements are capable of maintaining high performance in the life cycle with resource constraints, which is a tradeoff between user costs and agency costs. Actually, traffic distribution in a road network changes dynamically because of both the change of user costs caused by pavement performance deterioration and the decrease of traffic capacity brought about by maintenance actions. Traffic distribution can affect user costs in turn. In view of this, this research proposed a dynamic programming model combined with the stochastic user equilibrium model, which can simulate the dynamic traffic distribution in the life

cycle and developed a heuristic iterative algorithm to deal with the optimal model. The main results are shown as follows:

1.  The testing results prove that the proposed framework has an advantage in assessing user costs comprehensively and can provide an effective and optimal pavement maintenance strategy in a 30-year life cycle which improves the efficiency of budget and pavement conditions.
2.  We calculate the traffic flow, travel time and driving costs in the service stage and maintenance stage respectively in the 10th period for example. The results indicate that maintenance actions of five links divert the traffic flows by 11%, increase the average travel time by 44%, and increase the average driving costs by 31%, which provides quantitative evidence that interdependence exists in a road network. Interdependence should be accounted for in pavement maintenance decision-making.
3.  Sensitivity analysis reveals that there is a U-shaped relationship between $\rho$ and travel time as well as driving costs, which means that parameter $\rho$ has a great influence on the optimal decision making.

To our knowledge, few studies have yet focused on pavement maintenance optimization models considering dynamic traffic distribution or captured the interdependency in pavement maintenance optimization model. This research intends to contribute to the literature. However, there are two limitations in this research. First, we only focus on asphalt pavements in this research with no other types of pavements involved. Second, other uncertainties, such as the impact of traffic accidents on traffic distribution and climate change affecting pavement deterioration, should also be considered. The two limitations will be taken into account in the following studies.

**Author Contributions:** X.M. designed research methods and wrote the manuscript; C.Y. collected and analyzed the data and J.G. edited and revised the manuscript.

**Funding:** This research was funded by Humanities and Social Science Research Program of Ministry of Education in China (Grant Number 16XJCZH002) and supported by the Fundamental Research Funds for the Central Universities (Grant Number 310823170657 and 300102238501) and National Natural Science Foundation of China (Grant Number 71701022) and Natural Science Basic Research Plan in Shaanxi Province of China (Grant Number 2018JQ7002) and National Key R and D project (Grant Number 2017YFC0803906).

**Conflicts of Interest:** The authors declare no conflict of interest.

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
