# Peer review of "Incorporating Dynamic Traffic Distribution into Pavement Maintenance Optimization Model"

_sustainability, doi:10.3390/su11092488_

Round 1
Reviewer 1 Report
Interesting paper that provides a contribution on maintenance optimization operation of flexible pavements. From the revision, I would like to point out the following observations:
Assumption number 4 in section 3.1 is not clear, please explain better the meaning of “multiple roads” in the road network, since this may have several connotations.
A more detailed explanation of each formula should be provided when a specific reference is not indicated.
There are some English mistakes in the text. Please revise.
Author Response
Point 1: Interesting paper that provides a contribution on maintenance optimization operation of flexible pavements.
Response 1: The authors greatly appreciate the reviewer’s encouragement. This memo documents our responses to all review comments. The appropriate changes have been made to the manuscript.
Point 2: Assumption number 4 in section 3.1 is not clear, please explain better the meaning of “multiple roads” in the road network, since this may have several connotations.
Response 2: The authors have rewritten the assumption unmber 4 as “Maintenance actions shall be carried out for more than one road simultaneously instead of being one by one.”, which is clearer. Please refer to Line 126 and Line 127.
Point 3: A more detailed explanation of each formula should be provided when a specific reference is not indicated.
Response 3: The authors have provided a more detailed explanation for Equation (13). Please refer to Line 177 and Line 178.
Point 4: There are some English mistakes in the text. Please revise.
Response 4: The following revisions of grammar and spelling haven been done by authors.
Line 12: change “impact of changes……are rare to be investigated” to “impact of changes……is rare to be investigated”
Line 29: change “which needs periodic maintenance to ensure high level of performance” to
“which needs periodic maintenance actions to ensure high levels of performance”.
Line 34: change “prolong life apan of pavement” to “prolong pavements’ life span”.
Line 43: change “on the basis of user costs caused by pavement performance loss” to “on the
basis of pavement performance loss”
Line 49: change “The existing approaches dealing with pavement maintenance optimization” to “The existing approaches associated with pavement maintenance optimization”.
Line 69: change “identify the user costs generated in all stages” to “identify the life cycle user costs generated in all stages”.
Line 71 change “multiple time periods” to “every time period”.
Line 78: change “the allocation of resources” to “resource allocations”.
Line 79: change “maintenance of the pavements” to “maintenance actions of multiple pavements”.
Line 90-91: change “which is solved using stochastic programming” to “which was solved using stochastic programming models”.
Line 93: change “deal with” to “obtain”.
Line 94: change “maintenance problems” to “maintenance strategy”.
Line 102: change “solution methods” to “solution algorithms”.
Line 116: change “is developed to the model” to “is developed to solve the model”.
Line 128: change “by a maintenance action” to “after a maintenance action”.
Line 139: change “linkning” to “linking”.
Line 153: change “weight group” to “axle weight group”.
Line 223: change “Revise traffic volume” to “Adjust traffic volume”.
Line 231: change “get its optimal solution” to “denote its optimal solution”.
Line 239 change “with bituminous pavement” to “with bituminous pavements”.
Line 248: change “The budget of every year is constrained as US$ 45,000” to “The upper level of budget every year is US$ 45,000”.
Line 263: change “maintenance schedules” to “maintenance actions”.
Line 271: change “Link I is given pavement a maintenance action” to “Link I is given a pavement maintenance action”.
Line 319-321: change “make 11% of traffic flow divert their routes, generate a 44% increase in average travel time and a 31% increase in average driving costs” to “divert the traffic flows by 11%, increase the average travel time by 44% and increase the average driving costs by 31%”.
Reviewer 2 Report
Review of the paper entitled "Incorporating Dynamic Traffic Distribution into Pavement Maintenance Optimization Model" authored by Mao et al. (2019)
This research article focuses on optimizing pavement maintenance strategy using a multi-stage dynamic programming model combined with the stochastic user equilibrium model for simulating the dynamic traffic distribution in life cycle.
In my point of view, the authors study an interesting subject and I liked how they performed the simulations. However, to meet the high standards of the journal, the authors need to undertake moderate revision and improve the literature review. I would be happy to recommend this work for publication after seeing the comments are fully addressed.
Here is my list of comments/suggestions reported in what follows:
(1) How sensitive are the simulation results to the model parameters? In other words, did the authors perform any sensitivity analysis (at least one case as an example)?
(2) With regards to maintenance cost shown in Figure 6, what causes these fluctuations? The authors need to provide more analysis of these data.
(3) Could the authors account for other costs rather than the subject of this study? for example, for the transportation of pavement materials (asphalt and etc.)?
(4) Under what conditions, the model fails to provide accurate optimization?
(5) What other systems could be studied using this approach? Please provide some discussions on the applicability of this method to other systems.
(6) The quality of Figures 6-9 should be improved. Please improve the resolution.
(7) The manuscript requires proofreading. Few grammatical as well as typographical issues can be found in the manuscript.
(8) The literature review needs improvement and some experimental/modeling studies need to be mentioned with regards to pavement materials (asphalt...). The following studies can be considered: (i) Energy & Fuels, 29(9), pp.5595-5599. (ii) Colloids and Surfaces A: Physicochemical and Engineering Aspects, 513, pp.178-187. (iii) Physical Review E, 96(5), p.052803 (iv) Langmuir 33.8 (2017): 1927-1942.
Author Response
Point 1: In my point of view, the authors study an interesting subject and I liked how they performed the simulations. However, to meet the high standards of the journal, the authors need to undertake moderate revision and improve the literature review. I would be happy to recommend this work for publication after seeing the comments are fully addressed.
Response 1: The authors greatly appreciate the reviewer’s encouragement. The authors have made an appropriate revision to the manuscript.
Point 2: How sensitive are the simulation results to the model parameters? In other words, did the authors perform any sensitivity analysis (at least one case as an example)?
Response 2: The authors have added the sensitivity analysis of parameter into the manuscript. Please refer to the newly added section 5.3, Line 342- 338.
Point 3: With regards to maintenance cost shown in Figure 6, what causes these fluctuations? The authors need to provide more analysis of these data.
Response 3: The authors have analyzed the reasons of maintenance cost fluctuations among the 30 periods. Please refer to Line 278-280.
Point 4: Could the authors account for other costs rather than the subject of this study? for example, for the transportation of pavement materials (asphalt and etc.)?
Response 4: Maintenance costs in this research actually include material expense, transportation fee, labor cost, tax and so on. The authors have added the explaination. Please refer to Line 249-250.
Point 5: Under what conditions, the model fails to provide accurate optimization?
Response 5: The authors have added the explaination of the condition for optimization. Bellman’s principle of optimality is the condition of the optimization model. Please refer to Line 209-211.
Point 6: What other systems could be studied using this approach? Please provide some discussions on the applicability of this method to other systems.
Response 6: The authors have added the discussion on the applicability of this method to other systems into section 5.1. Please refer to Line 306-309.
Point 7: The quality of Figures 6-9 should be improved. Please improve the resolution.
Response 7: The authors have replaced original Figures 6-9 with new ones, which have higher resolution.
Point 8: The manuscript requires proofreading. Few grammatical as well as typographical issues can be found in the manuscript.
Response 8: The following revisions of grammar and spelling haven been done by authors.
Line 12: change “impact of changes……are rare to be investigated” to “impact of changes……is rare to be investigated”
Line 29: change “which needs periodic maintenance to ensure high level of performance” to
“which needs periodic maintenance actions to ensure high levels of performance”.
Line 34: change “prolong life apan of pavement” to “prolong pavements’ life span”.
Line 43: change “on the basis of user costs caused by pavement performance loss” to “on the
basis of pavement performance loss”
Line 49: change “The existing approaches dealing with pavement maintenance optimization” to “The existing approaches associated with pavement maintenance optimization”.
Line 69: change “identify the user costs generated in all stages” to “identify the life cycle user costs generated in all stages”.
Line 71 change “multiple time periods” to “every time period”.
Line 78: change “the allocation of resources” to “resource allocations”.
Line 79: change “maintenance of the pavements” to “maintenance actions of multiple pavements”.
Line 90-91: change “which is solved using stochastic programming” to “which was solved using stochastic programming models”.
Line 93: change “deal with” to “obtain”.
Line 94: change “maintenance problems” to “maintenance strategy”.
Line 102: change “solution methods” to “solution algorithms”.
Line 116: change “is developed to the model” to “is developed to solve the model”.
Line 128: change “by a maintenance action” to “after a maintenance action”.
Line 139: change “linkning” to “linking”.
Line 153: change “weight group” to “axle weight group”.
Line 223: change “Revise traffic volume” to “Adjust traffic volume”.
Line 231: change “get its optimal solution” to “denote its optimal solution”.
Line 239 change “with bituminous pavement” to “with bituminous pavements”.
Line 248: change “The budget of every year is constrained as US$ 45,000” to “The upper level of budget every year is US$ 45,000”.
Line 263: change “maintenance schedules” to “maintenance actions”.
Line 271: change “Link I is given pavement a maintenance action” to “Link I is given a pavement maintenance action”.
Line 319-321: change “make 11% of traffic flow divert their routes, generate a 44% increase in average travel time and a 31% increase in average driving costs” to “divert the traffic flows by 11%, increase the average travel time by 44% and increase the average driving costs by 31%”.
Point 9: The literature review needs improvement and some experimental/modeling studies need to be mentioned with regards to pavement materials (asphalt...). The following studies can be considered: (i) Energy & Fuels, 29(9), pp.5595-5599. (ii) Colloids and Surfaces A: Physicochemical and Engineering Aspects, 513, pp.178-187. (iii) Physical Review E, 96(5), p.052803 (iv) Langmuir 33.8 (2017): 1927-1942.
Response 9: It is a good suggestion, but since this manuscript focuses on asphalt pavement maintenance optimization model instead of asphalt material, the authors think its better to make literature review consistent with the research topic. But, we added some experimental/modeling studies of asphalt pavement from the literature suggested by reviewer into section 3.3. Please refer to Line 143-145.

Reviewer 3 Report
The manuscript is interesting and important from the science point of view, but the text is written with many shortenings and symbols. Maybe the authors can write the text more friendly to the readers when are used the words instead of symbols. But this is only the opinion, this is not a requirement.
Author Response
Point 1: The manuscript is interesting and important from the science point of view, but the text is written with many shortenings and symbols. Maybe the authors can write the text more friendly to the readers when are used the words instead of symbols. But this is only the opinion, this is not a requirement.
Response 1: The authors greatly appreciate the reviewer’s encouragement and suggestions. The authors have provided the explaination to every shortening and symbol.

Round 2
Reviewer 2 Report
I wanted to thank the authors for carefully addressing my comments. While I accept this paper for publication, please do the final minor comments for the sake of your nice article. Here are the final comments:
(i) Bold your findings, highlight the importance of this paper, and mention the future work.
(ii) Please do one more proofreading as I saw few grammatical issues.
Author Response
Point 1: I wanted to thank the authors for carefully addressing my comments. While I accept this paper for publication, please do the final minor comments for the sake of your nice article. Here are the final comments.
Response 1: The authors greatly appreciate the reviewer’s encouragement and suggestions. The appropriate changes have been made to the manuscript by the authors.
Point 2: Bold your findings, highlight the importance of this paper, and mention the future work.
Response 2: The authors have rewritten the Conclusions according to the reviewer’s comment. Please refer to Lines 396-425.
Point 3: Please do one more proofreading as I saw few grammatical issues.
Response 3: The following revisions of grammar and spelling haven been done by authors.
Line 10: change “in a high level” to “at a high level”.
Line 11: change “impact of changes in traffic distribution” to “the impact of changes in traffic distribution”.
Line 15: change “in life cycle” to “in the life cycle”.
Line 20: change “this research provides a quantitative evidence” to “this research provides quantitative evidence”.
Line 21: change “interdependency in road network” to “interdependency in a road network”.
Line 29: change “available budget” to “the available budget”.
Line 36: change “the tradeoff between user coasts and agency costs” into “the tradeoff between user cost and agency cost”.
Line 42: ” change “···maintenance actions, shown in Figure 1, where in a pavement life cycle···” to “···maintenance actions. Figure 1 shows that in a pavement life cycle···”.
Line 61: change “an stochastic dynamic process” to “a stochastic dynamic process”.
Line 77: change “Section 5 discusses key findings” to “Section 5 discusses the key findings”.
Line 82: change “an nonlinear programming model” to “a nonlinear programming model”.
Line 94: change “a MDP-based optimization model” to “an MDP-based optimization model”.
Line 102: change “algorithm[23] etc.” to “algorithm[23], etc.”
Line 105: change “algorithm[12] etc.” to “algorithm[12], etc.”
Line 107: change “vary” to “varies”.
Line 120: change “in life cycle” to “in the life cycle”.
Line 125: change “Pavement life cycle” to “The pavement life cycle”.
Line 116: change “··· , which affect users’ route choice” to “··· , which affects users’ route choice”
Line 138: change “traffic volume of link s” to “the traffic volume of link s”.
Line 154: change “Present Serviceability Index(PSI)” to “the Present Serviceability Index(PSI)”.
Line 155: change “by equivalent number of standard axle loads(ESALs)” to “by the equivalent number of standard axle loads(ESALs)”.
Line 162: change “3.4. Maintenance Cost Funcation” to “3.4. Maintenance Cost Function”.
Line 163: change “initial PSI level” to “the initial PSI level”.
Line 164: change “at the start of next period” to “at the start of the next period”.
Line 165: change “Life cycle is assumed” to “The life cycle is assumed”.
Line 168: change “in the beginning” to “at the beginning”.
Line 171: change “3.5. Use Cost Funcation” to “3.5. Use Cost Function”.
Line 176: change “in service stage” to “in the service stage”.
Line 201: change “left life time of link s” to “the left lifetime of link s”.
Line 201: change “the expected life time” to “the expected lifetime”.
Line 227: change “Bureau of Public Roads(BPR) function” to “the Bureau of Public Roads(BPR) function”
Line 270: change “···, which lead to a minimum PSI loss” to “···, which leads to a minimum PSI loss”
Line 272: change “but worst-first strategy” to “but the worst-first strategy”.
Line 273: change “best-first strategy” to “a best-first strategy”.
Line 310: change “Maintenance strategy” to “A maintenance strategy”.
Line 312: change “in 1st period” to “in the 1st period”.
Line 313: change “in 16th period” to “in the 16th period”.
Line 322: change “This research provides a quantitative evidence” to “This research provides quantitative evidence”.
Line 342: change “Lower value of ρ” to “A lower value of ρ”.
Line 343: change “While higher value of ρ” to “While a higher value of ρ”.
Line 351: change “a high performance” to “high performance”.
Line 352: change “user coasts” to “user costs”.
Line 357: change “in life cycle” to “in the life cycle”.
Line 365: change “provides a quantitative evidence” to “provides quantitative evidence”.
Line 371: change “such as impact of traffic accidents” to “such as the impact of traffic accidents”.
